# Optimizing the Design of Diatom Biosilica-Targeted Fusion Proteins in Biosensor Construction for *Bacillus anthracis* Detection

**DOI:** 10.3390/biology9010014

**Published:** 2020-01-07

**Authors:** Nicole R. Ford, Yijia Xiong, Karen A. Hecht, Thomas C. Squier, Gregory L. Rorrer, Guritno Roesijadi

**Affiliations:** 1Marine Biotechnology Group, Pacific Northwest National Laboratory, Sequim, WA 98382, USA; 2Department of Basic Medical Sciences, Western University of Health Sciences, Lebanon, OR 97355, USA; 3School of Chemical Biological and Environmental Engineering, Oregon State University, Corvallis, OR 97331, USA

**Keywords:** diatom, biosilica, biosensor, anthrax, biotechnology, molecular biology

## Abstract

In vivo functionalization of diatom biosilica frustules by genetic manipulation requires careful consideration of the overall structure and function of complex fusion proteins. Although we previously had transformed *Thalassiosira pseudonana* with constructs containing a single domain antibody (sdAb) raised against the *Bacillus anthracis* Sterne strain, which detected an epitope of the surface layer protein EA1 accessible in lysed spores, we initially were unsuccessful with constructs encoding a similar sdAb that detected an epitope of EA1 accessible in intact spores and vegetative cells. This discrepancy limited the usefulness of the system as an environmental biosensor for *B. anthracis*. We surmised that to create functional biosilica-localized biosensors with certain constructs, the biosilica targeting and protein trafficking functions of the biosilica-targeting peptide Sil3_T8_ had to be uncoupled. We found that retaining the ER trafficking sequence at the N-terminus and relocating the Sil3_T8_ targeting peptide to the C-terminus of the fusion protein resulted in successful detection of EA1 with both sdAbs. Homology modeling of antigen binding by the two sdAbs supported the hypothesis that the rescue of antigen binding in the previously dysfunctional sdAb was due to removal of steric hindrances between the antigen binding loops and the diatom biosilica for that particular sdAb.

## 1. Introduction

Diatoms are a group of unicellular microalgae, often with a highly silicified, mesoporous cell wall (frustule) exhibiting nano- to meso-scale hierarchical architecture [1,2,3]. Extraction of diatom frustules can be accomplished through acid washing (e.g., [4,5]) to remove all organic components, or by detergent extraction [6] to retain the protein complement. Acid washed diatom frustules resemble de novo assembled mesoporous silica nanoparticles [7,8]. Whether bioinspired or diatom-derived, these silicate structures can be chemically modified for a wide variety of uses, such as biomedical applications [9] (e.g., cell scaffolding [10,11] and drug delivery [12,13]) or biomass conversion [14]. One attractive method for functionalizing diatom biosilica is through genetic engineering of diatoms, whereby a biosilica-targeting protein or a derivative peptide thereof is fused to a protein of interest that is to be embedded into the biosilica through in vivo self-assembly. Such an approach does not require additional chemicals for attachment, and is scalable to large growth volumes.

The diatom *Thalassiosira pseudonana* is a model organism for in vivo self-assembly of genetically-modified frustules. As such, a number of proteins successfully have been used to functionalize the biosilica frustule of this diatom species [15,16,17,18]. Recent efforts in our lab focused on functionalization of the biosilica of *T. pseudonana* with chimeric fusion proteins consisting of the diatom-derived silica targeting peptide Sil3_T8_ [15,16] and a small synthetic antibody derivative (e.g., a single-chain variable fragment, scFv, or a single domain antibody, sdAb) [18,19]. Functionalization of *T. pseudonana* biosilica with an sdAb against the surface layer (S-layer) protein extractable antigen 1 (EA1) of *Bacillus anthracis* [18] by in vivo self-assembly utilized sdAb_EA1_, clone A1, which recognizes an epitope of EA1 accessible in lysed spores [20]. Preliminary work with sdAb_EA1_, clone G10, which recognizes an epitope of EA1 that is accessible in intact vegetative cells and spores [20], produced no functional diatom lines when targeted to the biosilica frustule. Noting that the N-terminus of the llama VHH domain sits adjacent to the binding loops (see Figure 2 of [21], for example), we hypothesized that our existing biosilica targeting constructs with the Sil3_T8_ peptide fused to the N-terminus of the sdAb_EA1_ might produce a fusion protein whereby the antigen’s access to the sdAb binding loops was occluded. For this particular protein to be functional when tethered to diatom biosilica, fusion protein structure needed to be optimized. Our solution was to uncouple the silica targeting peptide Sil3_T8_ from its ER targeting sequence and locate Sil3_T8_ at the C-terminus of the fusion protein. After doing so, the biosilica functionalized with sdAb_EA1_ clone G10—containing fusion proteins was able to bind its target antigen EGFP-tagged EA1 protein.

## 2. Materials and Methods

### 2.1. Diatoms

Native and transformed cultures of *T. pseudonana* (CCMP1335; Provasoli-Guillard National Center for Marine Algae and Microbiota, East Boothbay, ME, USA) were maintained in artificial seawater (ESAW; https://ncma.bigelow.org/media/pdf/NCMAalgalmedium.ESAW.pdf) supplemented with 100 μg/mL penicillin (VWR, Visalia, CA, USA) and 100 μg/mL streptomycin (Sigma-Aldrich, St. Louis, MO, USA) under continuous illumination on an orbital shaker (~10–40 μmol/m^2^/s, 20–22 °C) with gentle agitation or in a Caron (Marietta, OH, USA) plant incubator (150 μmol/m^2^/s, 20 °C) without agitation. Diatoms were transformed by microparticle bombardment with a PDS-1000/He particle delivery system as was previously described [18]. To verify integration of Gateway expression clones into the *T. pseudonana* genome, PCR was performed directly on 5 μL diatom culture (1/10 volume of PCR). Sequences of PCR primers are described in Appendix A. GAPDH was used as the control gene to verify presence of *T. pseudonana* DNA. Diatom biosilica frustules were isolated as previously described for detergent extraction at 50 °C with acetone rinse [6], but substituting 1% Igepal CA-630 (Sigma-Aldrich, St. Louis, MO, USA) for 1% SDS [18].

### 2.2. Expression Clone Construction

A diatom-specific destination vector (termed pDDV2 for Diatom Destination Vector #2) was created by restriction cloning to contain the *T. pseudonana fcp* promoter [22] followed by the endoplasmic reticulum trafficking sequence coding for the peptide MKTSAIVLLAVLATTAATEPR, the Gateway attR1/2 integration cassette with V5 and His6 tags, and the *T. pseudonana fcp* terminator [22]. The Multi-Site Gateway Pro cloning protocol was used to construct diatom-specific expression clones for biosilica-targeted fusion proteins using either the pDDV2 (this work) or the previously created pDDV1 [18]. Plasmids containing the unmodified sdAb_EA1_ (either clone A1 or G10) [20] and Sil3_T8_ [15,16] targeting sequences were used as templates for PCR to create entry clones for insertion into the pDDV vectors. Detailed descriptions of all clonings are available in the Appendix A.

### 2.3. Fluorescent Antigen Synthesis and Binding to Single Domain Antibodies

EA1-EGFP fusion protein was expressed and purified as previously described [18]. Isolated biosilica frustules from untransformed *T. pseudonana* and *T. pseudonana* cell lines transformed with various biosilica-targeted sdAb_EA1_ (either clone A1 or G10) fusion proteins were incubated with a saturating amount of EA1-EGFP (125 nM) in PBS containing 0.05% Tween-20 (Fisher, Hampton, NJ, USA) and 1% BSA Fraction V (Fisher, Hampton, NJ, USA) for 1 h at 4 °C, followed by 1 h at room temperature (20–25 °C). Antigen-bound frustules were washed three times with PBS containing 0.05% Tween-20 prior to imaging in the same buffer on PEI-coated coverslips. Frustule fluorescence was examined with a Leica DM IRB inverted epifluorescence microscope equipped with a mercury metal halide light source and liquid light guide (Leica, Wetzlar, Germany). A 460–500/505/512–542 nm filter cube was used to collect GFP fluorescence and a 635–675/716/696–736 nm filter cube was used to verify the absence of chlorophyll in frustule samples. Images were captured with a CoolSNAP Myo camera (Photometrics, Tucson, AZ, USA) and Metamorph software (v.7.7.11.0; Molecular Devices, San Jose, CA, USA). Frustules lacking detectable chlorophyll fluorescence and not overlapping any other frustules were manually selected and their GFP-channel fluorescence intensity measured using the Metamorph software package (v.7.7.11.0; Molecular Devices, San Jose, CA, USA).

### 2.4. Protein Modeling

The *T. pseudonana* silaffin precursor amino acid sequence (Sil3, GenBank: AAU44819.1) in FASTA format was input into SignalP-5.0 [23,24]. Eukarya was selected as the organism group, and long output format with figures was selected. SignalP-5.0 signal peptide prediction results are shown in Appendix A.

Homology modeling of the structures of the sdAbs was conducted using the SWISS-MODEL server [25]. The sdAb_EA1_ sequences (both clones A1 and G10) were uploaded to the ExPASy web server, and the server searched evolutionary related protein structures against the SWISSMODEL template library (SMTL) using two search methods: BLAST and HHblits. The templates were ranked according to expected quality of the resulting models, as estimated by Global Model Quality Estimate (GMQE) and Quaternary Structure Quality Estimate (QSQE). For each single domain antibody, the top-ranked template was chosen and the homology structure was built based on that template [25,26,27]. For sdAb_EA1_, clone G10, the top hit was 5F10 with a GMQE score of 0.78, and for sdAb_EA1_, clone A1, the top ranked template was 6GLW with a GMQE score of 0.77. The models were downloaded and the 3D protein structure visualizations and alignments were done with the open source protein structure visualization program Pymol [28]. Since neither of the top-ranked templates was a structure with bound antigen, we searched further in the template lists with lower GMQE scores to look for sdAb structures with bound antigen. Two protein structures were found: a fusion protein of two domains, Rpn8 and Rpn11, of the 26S proteasome’s deubiquitylation module bound to an sdAb (pdb bank ID 4OCN) [29] and Shiga toxin (stx2e) binding to a neutralizing sdAb (pdb bank ID 4P2C) [30] (Appendix A).

### 2.5. Statistical Analysis

The data were analyzed using one-way Analysis of Variance and Tukey’s HSD (honest significant difference) posthoc test for between-group differences (significant at *p* ≤ 0.05). The graphical comparisons in Figures 2 and 3 were derived from a study comparing diatoms transformed with generation 1 and generation 2, i.e., pDDV1 and 2 pDDV2, expression vectors- and a reference group comprising untransformed native diatoms.

## 3. Results

### 3.1. Re-Designing the Fusion Constructs

Previous reports that used Silaffin 3 (Sil3) [6] or its truncated derivative Sil3_T8_ [15] for in vivo tethering of fusion proteins to *T. pseudonana* diatom biosilica, located the silaffin domain at the N-terminus of the protein. When tethering an sdAb to the diatom biosilica in this fashion, we hypothesized based on structural considerations that the antigen binding loops would be oriented inward and adjacent to the biosilica rather than being exposed and facing outward away from the biosilica. In order to rotate the binding loops of our sdAbs away from the biosilica surface, and thereby increase antigen accessibility, the Sil3_T8_ tether needed to become a C-terminal fusion to the sdAb. But, because the Sil3_T8_ peptide contains an endoplasmic reticulum (ER) trafficking sequence [15,16], which needed to remain at the N-terminus of the fusion protein sequence, simply swapping the order of sdAb_EA1_ and Sil3_T8_ in our modular Gateway (Invitrogen) cloning system was not a feasible solution. Thus, re-designing the biosilica-targeting fusion constructs for retention of the ER trafficking sequence at the N-terminus of the peptide sequence was of primary concern.

Two changes were made to the ER trafficking sequence in relation to our previous Sil3_T8_ fusion constructs: the ER trafficking sequence was (1) lengthened and (2) relocated closer to the N-terminus of the fusion protein (Figure 1b). Analysis of the *T. pseudonana* Sil3 peptide sequence predicted that the original trafficking sequence of 17 amino acids ended at the cleavage site, and that the cleavage site may be more complex than previously assumed. Therefore, we increased the length of the trafficking sequence to include the first 21 amino acids of the Sil3 gene. By extending this peptide sequence we hoped to facilitate trafficking by allowing more efficient cleavage of the signal sequence.

Further, the Gateway (Invitrogen) modular cloning system leaves substantial scars between units. Our original design intended for the promoter to be one of the modular units. By leaving a scar after the promoter (which also contained the start codon for the fusion protein), the ER trafficking signal began 10 amino acids downstream from the N-terminus of the protein (Figure 1b, top). In order to continue to employ the modular Gateway (Invitrogen) cloning of our fusion partners, and use the scars as flexible linkers between fusion components, a new diatom-specific destination vector (pDDV2) was created with a static *fcp* promoter [22] that was immediately followed by the ER trafficking sequence. The Gateway (Invitrogen) scar would then be located downstream of the cleaved ER trafficking sequence (Figure 1b, bottom). Not only would this rearrangement of cloning elements allow the ER trafficking sequence to move closer to the N-terminus of the fusion protein (potentially allowing more efficient recognition), but it also uncoupled the remaining 37 amino acids of the Sil3_T8_ biosilica targeting peptide from the trafficking sequence so that the silica targeting peptide could be placed on either side of its fusion partner (Figure 1a).

In summary, the first generation diatom expression vector (pDDV1, Ref. [18]) was used to transform *T. pseudonana* resulting in the in vivo self-assembly of fusion proteins having a presumably suboptimal ER trafficking sequence (both in length and placement) and an N-terminal trafficking + targeting peptide/C-terminal sdAb orientation (Figure 1a, top). The second generation diatom expression vector (pDDV2) was used to transform *T. pseudonana*, resulting in the in vivo self-assembly of fusion proteins with the putatively improved ER trafficking sequence (both in length and placement), and the uncoupled targeting peptide Sil3_T8′_ located either on the N- (Sil3_T8′_-sdAb_EA1_; Figure 1a, middle) or C- (sdAb_EA1_-Sil3_T8′_; Figure 1a, bottom) terminus of the processed fusion protein.

### 3.2. Comparision of Two Generations of Fusion Constructs

Our earlier efforts to create an environmental biosensor for *B. anthracis* focused on the in vivo assembly of two sdAbs in diatom biosilica: sdAb_EA1_ clone G10, which binds intact spores, and sdAb_EA1_ clone A1, which binds lysed spores [20]. Given our previously inconsistent results in functionalizing diatom biosilica with these two sdAbs, we hoped that redesigning the expression vector would increase functionalization for both sdAbs. To that end, EA1-EGFP binding of three versions of each sdAb_EA1_ clone A1 (sdAb_EA1_/A1)–functionalized and clone G10 (sdAb_EA1_/G10)–functionalized diatom biosilica were compared (Figure 2 and Figure 3, respectively).

Considering sdAb_EA1_/A1 (Figure 2), the first generation pDDV1-derived Sil3_T8_-sdAb_EA1_/A1 functionalized biosilica frustules exhibited EA1-EGFP binding greater than that of native biosilica, similar to previous findings [18] in spite of the suboptimal ER trafficking sequence. Yet functionalized biosilica frustules with neither of the second generation pDDV2-derived constructs exhibited significantly improved EA1-EGFP binding in relation to the pDDV1-derived Sil3_T8_-sdAb_EA1_/A1 functionalized biosilica. These results indicated that the alterations to the ER trafficking signal sequence did not result in improved binding due to increased density of sdAb_EA1_/A1 in the frustule. The rotation of the sdAb in relation to the diatom biosilica also did not increase the amount of EA1-EGFP able to bind for this particular sdAb.

Considering sdAb_EA1_/G10 (Figure 3), the first generation pDDV1-derived Sil3_T8_-sdAb_EA1_/G10 in vivo functionalized biosilica frustules did not show EA1-EGFP fluorescence different from that of native biosilica, as expected from our previous observations. Conversely, in vivo functionalized biosilica frustules from both the second generation pDDV2-derived Sil3_T8′_-sdAb_EA1_/G10 and sdAb_EA1_/G10-Sil3_T8′_ exhibited significantly greater EA1-EGFP binding in comparison with the first generation pDDV1-derived construct. Although these findings do not exclude other factors associated with protein instability, they are consistent with results intended from the logical re-design of the constructs and pDDV2 to improve both ER trafficking and the release of steric hindrance at the antigen binding site.

### 3.3. Homology Modeling Suggests Differences in Antigen Binding Between sdAb_EA1_/A1 and sdAb_EA1_/G10

Based on sequence homology, the overall structures of both sdAb_EA1_/A1 and sdAb_EA1_/G10 are typical for sdAbs (Figure 4). This structure includes a heavy chain comprised of beta sheets, with N- and C-termini protruding from opposite ends of the 3-dimensional structure. The binding site, i.e., the complementarity determining region 3 (CDR3), is a loop structure (Figure 4, arrows). We predict the binding site for sdAb_EA1_/G10 to be located close to the N-terminus in the folded protein. Thus, N-terminal fusions with Sil3_T8_ would be expected to introduce steric hindrance that could prevent effective binding between the sdAb and the antigen. Antigen binding would not be disturbed if Sil3_T8_ was fused at the C-terminus of sdAb_EA1_/G10. We predict the binding site for sdAb_EA1_/A1, however, to be side facing, so that little steric hindrance would be introduced regardless of the orientation of the Sil3_T8_ peptide fusion with sdAb_EA1_/A1.

These models help to explain our observations in relation to sdAb_EA1_/A1, whereby EA1-EGFP binding is not substantially improved by our construct re-design. A side-facing binding surface would mean that sdAb_EA1_/A1 binding would not be sterically hindered by a fusion partner at either terminus of the peptide.

These models also supported our hypothesis that the primary reason that we did not observe EA1 binding by our first generation Sil3_T8_-sdAb_EA1_/G10-functionalized diatom biosilica was due to steric hindrance when the fusion protein was tethered to diatom biosilica by the N-terminally-located Sil3_T8_ silica-targeting peptide. Tethering to the diatom biosilica using the C-terminal fusion with the uncoupled Sil3_T8′_ biosilica-targeting peptide would have alleviated some of that stress.

## 4. Discussion

Individual protein structures often dictate the order of complex fusion protein partners. Problems arise when combining peptides with incompatible fusion preferences, as was found to be the case in creating a series of diatom biosilica-targeted constructs encoding sdAbs against the *B. anthracis* S-layer protein EA1. In particular, both the Sil3_T8_ (e.g., [12,15,16,32]) and sdAb_EA1_ (e.g., [33,34]) peptides previously have been made only as N-terminal fusions to their partners. One reason for this order of fusion partners is the requirement of many signal peptides’ proximity to the protein terminus, rather than just their amino acid sequence. Given our results with sdAb_EA1_/A1, some flexibility likely does exist with this placement (i.e., “close” to the N-terminus is good enough).

Further, the three-dimensional structure of each peptide in a chimeric protein must be considered. Even though much of the sdAb structure is conserved, the structure of the binding loops by necessity will vary by antigen and epitope recognized by the sdAb. For the two sdAbs described in this work, which bind unique epitopes in the same antigen, predicted structural differences of their respective binding loops may have caused the antigen to be positioned differently in relation to the static regions of the sdAb peptide, consequently altering the functionality of the sdAbs when they were tethered in diatom biosilica. This difference in binding location supports our observations whereby sdAb_EA1_/G10 is more sensitive to fusion partner orientation than sdAb_EA1_/A1. By uncoupling the trafficking and biosilica targeting domains of the Sil3_T8_ peptide sequence, we have increased the flexibility of using this particular peptide as a fusion partner for proteins that are more selective in regard to partner orientation.

In conclusion, we note that while modular cloning systems for synthetic biology are attractive due to their inherent convenience, cellular biology (e.g., protein trafficking) and biochemical (e.g., protein folding) considerations ultimately must guide the design of fusion proteins. This principle was especially relevant to the construction of functionalized diatom biosilica through genetic modification with two related sdAbs for the detection of the pathogenic bacteria *B. anthracis*.

## Figures and Tables

**Figure 1 biology-09-00014-f001:**
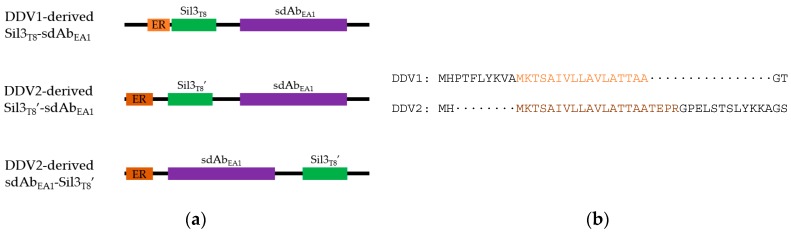
Diagrams of fusion proteins produced for this study. (**a**) Order of peptides in fusion proteins. Black lines—residual cloning scars; orange—ER trafficking sequence (light orange, 17 amino acids; dark orange, 21 amino acids); green—Sil3_T8_ biosilica-targeting peptide (Sil3_T8_, original composite peptide with ER trafficking sequence and silica targeting peptide; Sil3_T8′_, uncoupled silica-targeting peptide); purple—sdAb_EA1_ (can be either clone A1 or G10, exact identity is not specified here). Diagrams are for illustration purposes, and are not drawn to scale. (**b**) Alignment of ER trafficking sequences in each diatom expression vector. The sequence shown is from the N-terminus of the fusion protein through the second peptide feature of the protein. Colors are the same as in (**a**).

**Figure 2 biology-09-00014-f002:**
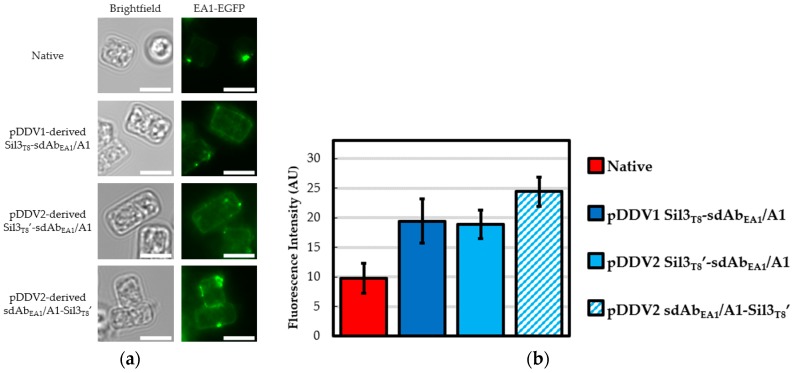
sdAb_EA1_/A1 in vivo functionalized biosilica. Diatom biosilica was isolated from native *T. pseudonana* and cell lines transformed with expression constructs containing various Sil3_T8_-sdAb_EA1_/A1 fusions. Presence of functional, in vivo assembled, biosilica-tethered sdAb_EA1_/A1 was confirmed by ability to bind EA1-EGFP protein. (**a**) Epifluorescence images of representative frustules showing different EA1-EGFP binding potentials. Given that S-layer proteins like extractable antigen 1 (EA1) (and, by extension, the EA1-EGFP used here) are self-assembling [31], the punctate fluorescence observed in all four panels is expected and contributes to the fluorescent signal observed for the native frustules. All calibration bars are 5 μm. (**b**) Average measured fluorescence intensities in arbitrary units (AU). Error bars are 95% confidence intervals (±t × 1 SEM). For native and pDDV1-derived Sil3_T8_-sdAb_EA1_/A1 biosilica frustules, n = 25 (df = 24, t = 2.06); pDDV2-derived Sil3_T8′_-sdAb_EA1_/A1 and sdAb_EA1_/A1-Sil3_T8′_ biosilica frustules, n = 100 (df = 99, t = 1.984).

**Figure 3 biology-09-00014-f003:**
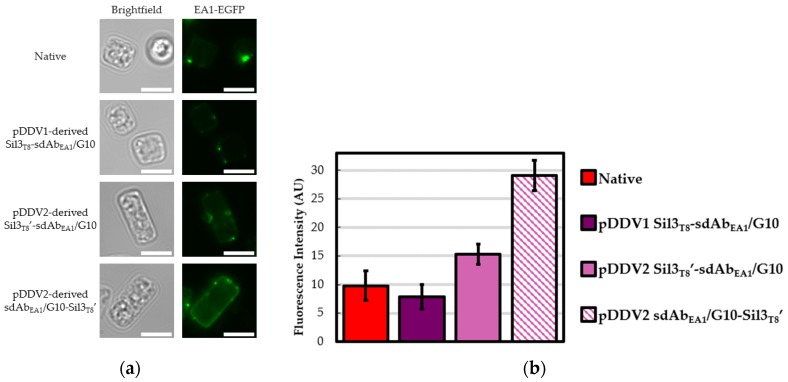
sdAb_EA1_/G10 in vivo functionalized biosilica. Diatom biosilica was isolated from native *T. pseudonana* and cell lines transformed with constructs containing various Sil3_T8_-sdAb_EA1_/G10 fusions. Presence of functional, in vivo assembled, biosilica-tethered sdAb_EA1_/G10 was confirmed by ability to bind EA1-EGFP protein. As all six single domain antibody (sdAb) fusions were tested in a single experiment, the native diatom biosilica results are identical to those presented in the previous figure. (**a**) Epifluorescence images of representative frustules showing different EA1-EGFP binding potentials. Given that S-layer proteins like EA1 (and, by extension, the EA1-EGFP used here) are self-assembling [31], the punctate fluorescence observed in all four panels is expected and contributes to the fluorescent signal observed for the native frustules. All calibration bars are 5 μm. (**b**) Average measured fluorescence intensities in arbitrary units (AU). Error bars are 95% confidence intervals (±t × 1 SEM). For native and pDDV1-derived Sil3_T8_-sdAb_EA1_/G10 biosilica frustules, n = 25 (df = 24, t = 2.06); pDDV2-derived Sil3_T8′_-sdAb_EA1_/G10 and sdAb_EA1_/G10-Sil3_T8′_ biosilica frustules, n = 100 (df = 99, t = 1.984).

**Figure 4 biology-09-00014-f004:**
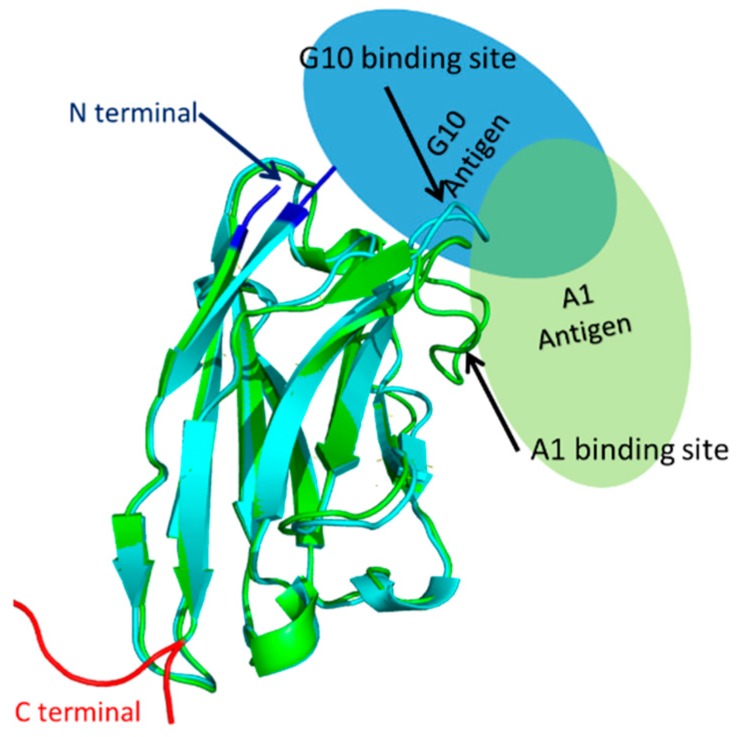
Aligned homology models of single domain antibodies with presumed antigen binding configuration. The ribbon diagrams of the homology structures of sdAb_EA1_/A1 (cyan, based on pdb ID 6GLW) and sdAb_EA1_/G10 (green, based on pdb ID 5F10) were aligned using Pymol. The ovals show the presumed binding positions of the EA1 antigen for sdAb_EA1_/A1 (light blue) or sdAb_EA1_/G10 (light green). The N- and C-terminal regions of the sdAbs are also noted in blue and red, respectively. See Appendix A for the individual pdb bank structures used to support the orientations of antigen binding.

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
