# Peer review of "Optimizing the Design of Diatom Biosilica-Targeted Fusion Proteins in Biosensor Construction for Bacillus anthracis Detection"

_biology, 2020, doi:10.3390/biology9010014_

Round 1

Reviewer 1 Report

In the manuscript “Optimizing the design of diatom biosilica-targeted fusion proteins in biosensor construction for Bacillus anthracis detection” Ford et al. describe how shuffling the order of peptides in diatom fusion proteins helps to avoid steric hindrances in the binding of antigen to a coupling of single domain antibody and biosilica targeting peptide Sil3T8.

The presented research is interesting, but the manuscript and its significance could be improved in several aspects:

> Introduction: The authors should explain in a sentence or two why they decided to use Thalassiosira pseudonana as a model diatom.

> Section 2.2.: When describing plasmid construction, it is customary to indicate which strain of Escherichia coli was used for cloning, along with its genotype.

> Section 2.3.: This section should contain a detailed description of the method used to measure fluorescence intensities in Figures 2b and 3b.

> Figure 2a/Native and Figure 3a/Native: The photograph appears to contain a circular artefact. A more representative photograph of negative control would be much preferable.

> As the main point of the manuscript is the design of functional fusion proteins in a diatom-specific destination vector, I would strongly recommend to include in the Supplementary materials complete sequence and annotation of plasmids pDDV2, pDDV2-sdAbEA1/G10-Sil3T8’ and pDDV2-sdAbEA1/A1-Sil3T8’  (e.g. in the textual GenBank format).

Author Response

We wish to thank all reviewers for their thoughtful critiques of our manuscript.  In response to all reviewer comments, we have altered our manuscript. We also would like to note several substantial modifications that we have made:

We have decided to reverse the order of Figures 2 and 3, as we feel doing so improves the flow of our manuscript. We have changed the error bars on Figures 2b and 3b to be 95% confidence intervals rather than the SEM. We have substantially modified the language in section 3.3. We have sought to more clearly emphasize that the homology modeling is only a prediction that supports our data, and not reflective of any structural knowledge of our fusion constructs.

In response to the reviewer’s five specific improvements:

In the Introduction, the reviewer requested that we describe why we chose the diatom species pseudonana. We have added a sentence at the beginning of the second paragraph (line 48) to emphasize that T. pseudonana has been used previously by us and by others to produce in vivo functionalized biosilica frustules. We chose the species because the system for transformation and integration of fusion constructs had been established already. The reviewer requested that we provide information about the strains of coli used for cloning and expression purposes. As neither the propagation nor expression (when applicable) of fusion constructs required any special manipulations, the strains have been described as the last section of the Supplementary Methods. The reviewer requested more information about how fluorescence intensity values were obtained. Section 2.3 (begins on line 93) now contains additional details pertaining to fluorescence intensity measurements. The reviewer requested that the control image in Figures 2 and 3 be replaced. The circular artifact that inspired this request in Figures 2 and 3 is not an artifact. Rather, it is a valve-view of a frustule.  Showing both the girdle (side)- and valve (top)-views of diatom biosilica frustules was intentional on the part of the authors, and the image has not been changed. The reviewer requested that we provide the sequences of the fusion proteins in the Supplementary materials, and we have done so. This section can be found between the Supplementary Figures and Supplementary References.

Reviewer 2 Report

The manuscript by Ford et al describes a strategy to functionalize biosilica frustules from diatom algae with fusion proteins to serve as antibody-based biosensors. Fusion proteins composed of single domain antibodies with a silica targeting peptide can be expressed in the algae and the biosilica scaffold with bound proteins can be obtained after extraction. For unknown reasons, a particular antibody failed to bind its antigen which triggered the authors to develop strategies to either improve targeting or relieve sterical hinderance within the fusion protein. The approach and the resulting strategies can be of value to future studies aiming at immobilizing fusion proteins at diatom biosilica frustules. Unfortunately, while developing sound hypothesizes and eventually obtaining a functional fusion protein, the authors did not discover what the original problem was that rendered the fusion protein inactive. It could have been any of the following: low expression, poor targeting to a) ER or b) silica, misfolding, degradation or, as the authors describe, sterical hinderance due to the N-terminal tag. Resolving the origin of the initial failure would have been of great interest and enable future researchers intelligent design of such fusion proteins. Nevertheless, this manuscript can serve as inspiration how to approach similar problems.

Major comments:

The manuscript deals with several cloning problems that lead to potentially dysfunctional fusion proteins. These problems originate in use of an outdated cloning system, Gateway. These problems could have been avoided by use of modern cloning techniques such as Gibson assembly which is also restriction enzyme independent and modular but does not leave ‘scars’. The company Invitrogen that developed this system does not exist any more since 2008 but is now a brand of ThermoFisher. What is the origin of failure to functionalize biosilica with the original fusion protein? Is it a) expression levels, b) ER targeting, b) biosilica targeting or c) conformational/sterical hinderance? It would be valuable to distinguish between these very different potential bottlenecks. At least expression levels and biosilica targeting can be investigated by means of western blot analysis. It should be possible to extract biosilica adsorbed protein to test how much of the various sdAb is attached to the biosilica. In Figure 3, the sdAb constructs without silica targeting peptide bind very efficiently to the biosilica. Why? Presumably, the A1 and G10 antibodies are not vastly different in their general properties, why does one stick to the biosilica and the other not? Compare Figure 2 and 3 the middle panels, pDDV1 and pDDV2 constructs without Sil3. What is the origin of this difference? The constructs without Sil3 should be soluble and be extracted. Please provide a control where you show that you efficiently extracted the cytoplasm and membranes of the diatom. It is unclear whether the experiments were done with cleanly extracted biosilica.

Minor comments:

Please explain briefly how a sdAB-decorated biosilica frustule could be used as a biosensor. The names of the antibodies and fustion protein constructs is very long and confusing also because the main difference is in the subscript letter (A1 vs G10, what does that mean?). In order to not loose the reader, please rename into something simple. How was the fluorescent intensity quantified? Please perform statistical analysis. This is especially important for the data in figure 3. Possibly, the difference between the constructs without and with Sil3 is not significant. Line 202: Steric hinderance is also not important considering only a marginal increase in EA1-capture between the two DDV2 constructs. Line 223: Please add reference and describe the study that showed the binding sites. Fig 4 legend: Please add PDB code and reference here or in the main text, not only in supplemental info.

Author Response

We wish to thank all reviewers for their thoughtful critiques of our manuscript.  In response to all reviewer comments, we have altered our manuscript. We also would like to note several substantial modifications that we have made:

We have decided to reverse the order of Figures 2 and 3, as we feel doing so improves the flow of our manuscript. We have changed the error bars on Figures 2b and 3b to be 95% confidence intervals rather than the SEM. We have substantially modified the language in section 3.3. We have sought to more clearly emphasize that the homology modeling is only a prediction that supports our data, and not reflective of any structural knowledge of our fusion constructs.

We appreciate the reviewer’s critical analysis of our manuscript. We address the reviewer’s comments below:

The reviewer indicated that our diatom expression constructs could have been more suitably designed using a scarless cloning method. We respectfully submit that choice of a cloning method which produced scars between modules was intentional. In the creation of our complex fusion proteins, we wanted to include an unstructured linker. To that end, the Gateway scars were desirable in that they became that separation between protein domains. Therefore, we have included a clause to emphasize this positive aspect to our cloning preference (Lines 156-157). Additionally, we wish to note that an important message of our manuscript is that no single cloning scheme will be perfectly suited to every situation. What we have presented here is one possible solution to that problem, which required a minimum amount of change for our group’s cloning strategy of choice.

We suspect that the reviewer mis-interpreted our use of the term “functionalized biosilica” to mean that our fusion constructs were expressed and purified outside of our diatoms, then adsorbed after biosilica extraction. Such was not the case. We cannot fully address the reviewers concerns that follow this line of logic, as they do not apply to our system: All of our sdAb-containing fusion proteins were integrated into the diatom genome to produce stable diatom lines that expressed the biosilica-targeted fusion proteins. To reiterate, all fusion proteins contained the silica-targeting peptide. We have attempted to emphasize that we used “in vivo self-assembly” as much as possible to prevent future readers from encountering the same misunderstanding.

The reviewer requested that we rename our fusion constructs. While we understand that our complex fusions have complex names, our intent was to keep our terminology as consistent with past publications as possible. To help readers navigate the manuscript, we have made an effort to use more descriptive language in addition to the proper names of the fusion proteins. We have also reduced the amount of subscript used, so as to draw readers’ attention to the unique features of our constructs. Additionally, we formalize the abbreviations for the clone A1/G10 terminology on line 189 of the manuscript, and the Sil3T8’ peptide on Lines 169 and 180 of the manuscript.

The reviewer requested more details about the fluorescence intensity measurements and analysis.  Section 2.3 (begins on line 93) now contains additional details pertaining to fluorescence intensity measurements. As previously noted, we now have elected to present our data graphically using the 95% confidence intervals rather than the SEM.

In response to the reviewer’s specific line item requests, we note:

Regarding line 202 in the original text, the reviewer noted that the results presented from clone A1-functionalized diatom biosilica should be interpreted as refuting our conclusion that steric hinderance was the reason our initial work with clone G10 was unsuccessful. We respectfully point the reviewer to Section 3.3 of the manuscript where we suggest that that clone A1 may not be subject to the same amount of steric hindrance as clone G10 due to a potential difference in the locations of the antigen binding sites. Regarding line 223 in the original text, the reviewer requested that we add a reference for how the binding sites were determined. We have re-worded this section to clarify that homology modeling was performed and the limits to this analysis. References for the homology modeling method are formally cited in the Methods section of the manuscript (Section 2.4). The reviewer requested that we add the pdb codes for the homology models to the Figure 4 legend. We have done so. They can be found on line 258 of the revised text.

Finally, we agree completely with the reviewer that additional avenues related to protein stability and expression could be explored in the future. To that end, we have stated formally in our revised manuscript in the sentence that begins on line 217 that other factors may contribute to a failure to produce functional fusion proteins. We also note that due to changes in author employment status, we do not have any further experimental work to include in the manuscript. As we did not wish to withhold what data we did have from the community, we chose to use a Short Communication format to disseminate our findings. We hope that publication of our work in its current form will inspire our peers to find equally creative solutions to their problems and help us complete our story.

Reviewer 3 Report

This short communication focuses on optimizing fusion proteins targeted to improving their function as biosensor for Bacillus anthracis detection. The authors modified their previous constructs by increasing the length of ER signal sequence of Sil3T8 and swapping the rest of Sil3T8 with the sdAbEA1 sequence. They found that the resulted sdABEA1/G10-Sil3T8’ fusion protein on diatom frustules exhibited significantly stronger binding capacity with EA1-EGFP antigen than Sil3T8’-sdABEA1/G10, as well as the previous fusion protein Sil3T8-sdABEA1/G10. In contrast, by using the same strategy, the resulted sdABEA1/A1-Sil3T8’ fusion protein showed similar binding activity as Sil3T8’-sdABEA1/A1 and the previous Sil3T8-sdABEA1/A1. With the help from homology modeling, they concluded that the steric hindrance caused by the arrangement of fusion protein, rather than the length of ER signal sequence, was the main reason that led to dysfunction of sdAbEA1/G10 in fusion protein Sil3T8-sdABEA1/G10. This is a neat and straightforward piece of work. The manuscript was well organized and written.

Concerns/suggestions:

Considering the steric hindrance, swapping the domains of the fusion protein (while keep the ER signal sequence) is one solution. Another potential approach is to add a relative long and flexible linker sequence between the Sil3T8 and sdAbEA1 domains in Sil3T8-sdABEA1/G10. The steric hindrance may be due to the sdAbEA1 domain was too close to the cell wall of diatom, which was resulted from binding of Sil3T8 to biosilica on frustules and subsequent drag of the sdABEA1 domain close to the cell wall. Without crystal structure, modeling can only be deemed as side evidence. For the homology modeling analysis in this manuscript, it is better to cite previous literature that used similar approach. In Sil3, the ER signal sequence is MKTSAIALL…. In figure 1b, the sequence is MKTSAIVLL…. In figure legend of Fig 1, it is better to add a description for Sil3T8’, to distinguish the difference from Sil3T8. In figure legend of Fig 2, the reviewer suggests adding a sentence to explain the presence of dot fluorescence in the native panel. In Fig 3b, from the error bars, the fluorescence intensity of pDDV2-sdAbEA1/A1-Sil3T8’ looks significantly higher than the others. This is consistent with the result showing in Fig 3a. Is it true? How to explain it?

Author Response

We wish to thank all reviewers for their thoughtful critiques of our manuscript.  In response to all reviewer comments, we have altered our manuscript. We also would like to note several substantial modifications that we have made:

We have decided to reverse the order of Figures 2 and 3, as we feel doing so improves the flow of our manuscript. We have changed the error bars on Figures 2b and 3b to be 95% confidence intervals rather than the SEM. We have substantially modified the language in section 3.3. We have sought to more clearly emphasize that the homology modeling is only a prediction that supports our data, and not reflective of any structural knowledge of our fusion constructs.

In response to the reviewer’s concerns:

The reviewer noted that our solution to the problem was not unique. We agree, and have modified the emphasis at the end of the Introduction (line 63 onward) to emphasize that fact. We would note that the reviewer’s solution to increase the linker length is a good solution, but was not considered in our initial strategy due to our observations (also published in our previous work – see Ref. 8 from our manuscript) that longer, more complex fusion proteins were not expressed by our diatoms. To elaborate, fusion proteins containing the silica targeting peptide with a GFP-tagged sdAb were not expressed successfully in diatoms, but equivalent versions without the GFP tag were expressed successfully. Therefore, we desired a solution that would not substantially increase the size of our fusion protein.

The reviewer noted that homology modeling should only be deemed as side evidence. We agree, and as we stated above, we have altered Section 3.3 (begins on line 242) to be more conscientious of our assertions.

The reviewer noted that our ER trafficking sequence contains a single amino acid change from the Sil3 ER trafficking sequence. In the interest of accurately displaying our data we wish to leave Figure 1b as displayed. The sequence presented is the sequence that was amplified from the original Sil3T8-containing plasmid obtained from the authors of Ref. 15 (as noted in the acknowledgements of our manuscript). Sequencing of this plasmid confirms that the sequence presented in Figure 1b is the sequence that was present in our template stock.

The reviewer requested that we more clearly differentiate Sil3T8 and Sil3T8’ peptides. We have noted the distinction in the legend to Figure 1 (see line 169). We also restate the difference in the main text (see line 180). We also have chosen to remove as much information in the construct names as possible from subscripts to increase clarity.

The reviewer suggested that we should address the presence of fluorescence in our negative control for Figures 2 and 3. We have done so in the respective figure legends as requested (see lines 205-207 and lines 227-229). Briefly, the many bright fluorescent spots seen throughout Figures 2 and 3 are due to the self-assembling nature of the antigen protein.

Round 2

Reviewer 2 Report

While some of my previous comments and requests for additional experiments could not be addressed due to change in author employment, the revised manuscript contains clarifications and improvements. I would like to encourage the authors to investigate what the original problem was so that future studies can be guided by the results.